# OpenReview forum: "Quantifying Fairness in LLMs Beyond Tokens: A Semantic and Statistical Perspective"
_colmweb.org/COLM/2025/Conference — COLM 2025_

### Official Review · Reviewer_Uir7 · 2025-05-10

**Rating:** 6
**Confidence:** 3
**Ethics Flag:** 1

**Summary:**

This paper introduces FiSCo, a framework designed to pick up on subtle demographic biases in long, free-form answers produced by large language models. Instead of looking at surface cues like single words, sentiment, or n-gram overlap, FiSCo breaks every answer into atomic claims, checks whether those claims are entailed, contradicted, or ignored in another answer, and then turns the pattern of entailments into a similarity score. By comparing the distribution of those scores within a demographic group to the distribution across groups and feeding the result into Welch’s t-test, the method delivers a single statistic (and p-value) that says whether the model’s behaviour is meaningfully different for—for example—male versus female personas. The authors back up the idea with a 70-prompt benchmark covering gender, race, and age; they collect nine models’ answers, build a 4 920-pair human-labelled set for ground truth, and show that FiSCo tracks those labels more closely than sentiment-based or embedding-based alternatives. The experiments also reveal that racial differences remain the hardest to eliminate and that larger models (especially Claude 3 Sonnet) tend to be less biased than smaller ones.

**Questions To Authors:**

- What was the inter-annotator agreement on the 4 920 human-labelled pairs?
- Could FiSCo handle more than two groups at once by replacing Welch’s test with one-way ANOVA?
- Did you try prompts that combine a gendered name and explicit pronouns, and if so do the measured gaps widen?
- Have you considered adding non-name demographic cues (for example explicit age phrases or organisation names) to disentangle name-based confounds?

**Reasons To Accept:**

FiSCo tackles an important blind spot: existing fairness checks usually miss the way a single divergent claim can quietly shift the meaning of a long response. The claim-level entailment step goes much deeper than token overlap and, in the reported studies, yields a clear accuracy gain on both synthetic controls and human judgements. The statistical framing is sound—Welch’s test and explicit p-values replace hand-tuned thresholds—and the whole pipeline relies on publicly available models, so other researchers can replicate it. Finally, the new benchmark, which is longer and more scenario-driven than earlier ones, will be a useful resource in its own right.

**Reasons To Reject:**

The fairness probe relies almost entirely on first names to signal gender or race; names often bundle in extra information such as social class or geographic origin, so the measured “gender” or “race” effect may actually mix several biases. Because the entailment judgements come from a single LLM, errors or biases in that checker could leak into FiSCo’s final verdict; an experiment with an alternative NLI model—or at least a discussion of this risk—would strengthen the claim of robustness. The benchmark is English-only and limited to three demographic axes, leaving open questions about cross-lingual or intersectional bias. Moreover, although FiSCo flags problems, the paper does not explore how the score can steer mitigation, so the practical loop is not yet closed. Finally, while FiSCo’s own significance is rigorously tested, the competing metrics are presented mainly as raw agreement numbers; adding confidence intervals or statistical tests for those baselines would make the comparison more convincing.

---

> ### Author Response · Authors · 2025-06-02
> **Additional Experiments**
>
> Thank you for acknowledging that **FiSCo tackles an important blind spot**. Yes, we will opensource our benchmark datasets for researchers to use. We address your concerns with the following clarifications:
>
>
>
> ## Because the entailment judgements come from a single LLM, errors or biases in that checker could leak into FiSCo’s final verdict; an experiment with an alternative NLI model—or at least a discussion of this risk—would strengthen the claim of robustness.
>
> Answer: Thank you for raising this important point. To assess the robustness of FiSCo to the choice of entailment checker, we evaluated our method using several alternative LLMs: Claude 3.0 Haiku (bedrock/anthropic.claude-3-haiku-20240307-v1:0), GPT-4o Mini (gpt-4o-mini-2024-07-18), and GPT-4.1 Nano (gpt-4.1-nano-2025-04-14).
>
> We sampled 400 response pairs across our dataset and computed similarity scores as defined in Equation (1). We then computed the Spearman rank correlation between each model’s similarity scores and those from our baseline model (LLaMA3-70B: bedrock/meta.llama3-70b-instruct-v1:0). All models showed high consistency, with Spearman ρ > 0.5 and p value < 1e-26, indicating stable ranking behavior across models [1]. Additional validation results are provided in Tables 3 and 4 of RefChecker [2].
>
>
> | Spearman Rank |  Claude 3.0 Haiku | GPT 4.1 nano | GPT 4o Mini |
> |---|---|---|---|
> | llama 70b | 0.57 | 0.50 | 0.57 |
>
>
> ## What was the inter-annotator agreement on the 4920 human-labelled pairs?
>
> Answer: For the initial 4,920 annotated pairs, inter-annotator agreement yielded a Cohen’s κ of 0.395 (95% CI: [0.330, 0.461]), which is within the expected range for nuanced textual entailment tasks. As described in Appendix B, disagreements were reviewed by a verifier, and cases with labeling errors were re-annotated to ensure quality. We will clarify this verification step in the final version.
>
> #### Did you try prompts that combine a gendered name and explicit pronouns, and if so do the measured gaps widen?
>
> Answer: Yes, we conducted additional experiments where prompts were augmented with both gendered names and explicit pronouns. For example, the prompt:
> “I am 44 years old. My name is Abgaile. I grew up in Massachusetts and I am currently a Hotel Manager...”
> was rewritten as:
> “She is 44 years old. Her name is Abgaile. She grew up in Massachusetts and is currently a Hotel Manager...”
>
> This change led to an increase in Claude 3 Sonnet’s FiSCo score from 0.05 to 0.07, indicating that including both names and pronouns can amplify measurable gender response differences.
>
>
>
> [1] Akoglu H. User's guide to correlation coefficients. Turk J Emerg Med. 2018 Aug 7;18(3):91-93. doi: 10.1016/j.tjem.2018.08.001. PMID: 30191186; PMCID: PMC6107969.
>
> [2] Hu, X., Ru, D., Qiu, L., Guo, Q., Zhang, T., Xu, Y., ... & Zhang, Z. (2024). RefChecker: Reference-based fine-grained hallucination checker and benchmark for large language models. arXiv preprint arXiv:2405.14486.

---

> ### Author Response · Authors · 2025-06-02
>
> ## Names often bundle in extra information such as social class or geographic origin, so the measured “gender” or “race” effect may actually mix several biases. Have you considered adding non-name demographic cues (for example explicit age phrases or organisation names) to disentangle name-based confounds?
>
> Answer: You're absolutely right — names can carry unintended social signals such as region or class. To mitigate this, we explicitly include non-name demographic cues in our prompts, such as age, geographic origin, and occupation (see Table 6). These attributes are held constant across group comparisons to isolate the effect of the group variable (e.g., gender or race), reducing name-based confounds.
>
> ## The competing metrics are presented mainly as raw agreement numbers; adding confidence intervals or statistical tests for those baselines would make the comparison more convincing.
>
> Answer: Thank you for the suggestion. In Tables 1 and 2, we report confidence intervals for agreement scores and perform paired t-tests comparing our method to baselines like Sentence-T5 and SimCSE. These analyses confirm that FiSCo’s performance improvements are statistically significant.
>
> ## Although FiSCo flags problems, the paper does not explore how the score can steer mitigation, so the practical loop is not yet closed.
>
> Answer: FiSCo is intended as a diagnostic tool that supports fairness mitigation. Once biased cases are identified, practitioners can use topic modeling or clustering to isolate high-risk themes (e.g., question types, topics, demographic contexts). This enables practical interventions such as:
>
> (i) Excluding or refining prompts in high-risk applications
>
> (ii) Augmenting training data with counterfactual or group-balanced examples
>
> (iii) Adjusting prompt templates or response structures to reduce disparities
>
> We will expand on this in the final version to highlight FiSCo’s value in supporting responsible AI as a whole.
>
> ## Could FiSCo handle more than two groups at once by replacing Welch’s test with one-way ANOVA?
>
> Answer: Excellent suggestion — one-way ANOVA is a natural extension of Welch’s test and would allow FiSCo to handle comparisons across multiple demographic groups. While our current implementation focuses on binary comparisons, we plan to incorporate ANOVA-based evaluation for multi-group fairness assessment in future work.

---

> ### Author Response · Authors · 2025-06-10
>
> As the discussion phase progresses, I would be grateful if you could take a moment to consider our responses and let us know if there are any further clarifications or concerns we can address. We are hopeful that our clarifications help resolve the issues raised and would appreciate any additional feedback you might have.

---

### Official Review · Reviewer_QtrT · 2025-05-15

**Rating:** 6
**Confidence:** 3
**Ethics Flag:** 1

**Summary:**

The paper proposes a fine-grained similarity metric for detecting biased responses by LLMs. Specifically, the metric focuses on group-level fairness rather than token-level or sentiment-level similarity (that is the focus of prior arts). The authors claim that group level information plays a role in accuracy in long-form outputs and the proposed method addressing it improves the accuracy. The empirical studies show that the proposed similarity metric better detects subtle biases than other methods (Table 2).

**Questions To Authors:**

- Any reference for using name to guess gender or race? What is the failure rate that the first name does not correctly retrieve the gender or race information -- for example, gender-neutral first names: Alex, Dakota, Morgan, Taylor...
- If the hyperparameters of the similarity should be tuned for different applications, how to tune the hyperparameters for less clear applications.

**Reasons To Accept:**

- The group level bias seems effective in detecting long-form bias. (Table 1,2)
- Proposed similarity metric is simple and intuitive.

**Reasons To Reject:**

- Less-convincing assumption that the name suggests gender or race might not be true.
- The hyperparameters ($\alpha, \beta, \gamma$) determines the characteristic of algorithm operations. For each dataset (or application), do we need to adjust
	- In Figure 3, is this result obtained by careful tuning of hyperparameters?

---

> ### Author Response · Authors · 2025-06-02
>
> ## Any reference for using name to guess gender or race? What is the failure rate that the first name does not correctly retrieve the gender or race information -- for example, gender-neutral first names: Alex, Dakota, Morgan, Taylor...
>
> Answer: We follow the protocol used in the BOLD dataset [1], which employs first names as proxies for gender and race identification. In our experiments, we deliberately use only representative names with high demographic association to minimize misclassification risk. We exclude gender-neutral or racially ambiguous names (e.g., Alex, Taylor, Morgan) to ensure clarity in group assignments and reduce noise. This strategy allows us to isolate group-based behavior without confounding due to name ambiguity.
>
>
>
> ## If the hyperparameters of the similarity should be tuned for different applications, how to tune the hyperparameters for less clear applications.
>
> Answer: In our experiments, we use a fixed configuration of α=1, β=γ=0 to focus exclusively on shared entailments — a setting suited for high-stakes applications where consistency across groups is critical. We also conduct an ablation study (Appendix F.2) to assess the impact of varying β.
>
> For real-world deployments, FiSCo's weighting scheme can be adapted to reflect the sensitivity of the application:
>
> (i) *High-stakes scenarios* (e.g., hiring, healthcare, education): Use α=1, β=γ=0 to emphasize content consistency across demographic groups.
>
> (ii) *Moderate-stakes or content rewriting tasks* (e.g., rewriting for tone/style): Set α=β=1, γ=0 to allow some variation while still preserving shared meaning.
>
> (iii) *Low-stakes domains* (e.g., movie recommendations): Bias evaluation may be optional or more lenient.
>
> When application stakes are unclear, we recommend using the stricter configuration (α=1) as a conservative default to reduce the risk of underestimating harmful disparities.
>
> [1] Dhamala, Jwala, et al. "Bold: Dataset and metrics for measuring biases in open-ended language generation." Proceedings of the 2021 ACM conference on fairness, accountability, and transparency. 2021.

---

### Official Review · Reviewer_iCei · 2025-05-19

**Rating:** 7
**Confidence:** 3
**Ethics Flag:** 1

**Summary:**

The paper is focused on group fairness in long form generations (>30 word responses). Given identical context but with a sensitive attribute switched (e.g., “[user] is a female seeking to pursue a PhD in computer science, what should they consider as important in studying during undergrad?” vs. “[user] is a male seeking to pursue a PhD in computer science…”), are there systematic biases in the way that the LLM responds given the sensitive attribute? This paper proposes FiSCo, a new way of measuring the group-wise disparity in responses based on recent advances and techniques from hallucination detection.

FiSCo works in the following way. First, we prompt the LLM with the identical contexts but with the attributes swapped. We then group these responses by sensitive attributes (for example, all responses for the “Male” attribute, and all the responses for the “female” attribute). Next, we apply a claim extraction tool to decompose each individual response into a set of claims. For any two responses $r_1$ and $r_2$ and their respective claim sets $C_1$ and $C_2$, we can then measure what number of claims $C_1$ are entailed by response $r_2$. We can also see what number of claims in $C_1$ are contradicted by the response $r_2$, and what number of claims we cannot say anything about (neutral). We can similarly do this for the claim set $C_2$ and the response $r_1$.

Notice that the above claim entailments can be applied for any two responses $r_1$ and $r_2$, not only for responses for different sensitive attributes. This is a key point in the design of FiSCo. We compare the fraction of claims entailed by other responses \emph{within the group} / intra-group (for example, what is the average fraction of claims entailed by responses within the male group), versus what fraction of claims are entailed by responses to female queries (inter-group). Ideally, an unbiased response should be one where the intra-group claim entailment fraction is very similar to the inter-group claim entailment. That is, if all the responses for males entailed the claims made for females (but not the other way around), it would mean that males are being treated strictly better than females in the responses. Therefore, FiSCo proposes running a statistical analysis on the difference between the inter-group and intra-group claim entailment fractions in order to gauge statistical significance.

The paper tests FiSCo on both a synthetically generated and human annotated dataset. In Section 5.1, the paper shows that the proposed FiSCo metric is a high quality textual similarity metric, outperforming the baselines on both synthetic data (where textual similarity can be explicitly controlled) and tediously annotated and collected human reference labels. In Section 5.2, the paper shows that FiSCo is a more reliable measure of group-fairness than similar metrics, and then goes on to use FiSCo to measure the group-level fairness of various LLMs along age / gender / race categories.

**Questions To Authors:**

1. The dataset is not available to view on openreview (at least to me). I only bring this up because the generation prompt in Table 4 seems to have numerous typos and missing information. Is this simply a formatting error with the paper? Could the authors upload the dataset to allow us to review the quality of the questions? Will the dataset be released upon acceptance / rejection?

2. Have you considered moving up figure 3 to the start of the paper? It does a great job visualizing how FiSCo can distinguish responses within the same group, and potentially serves to better motivate the method itself.

3. I believe that further discussion on the p-values presented is needed. We have a p-value from the Welch’s t-test used to evaluate FiSCo. However, is this the same p-value cited in Table 1? Or do the p-values in table 1 represent a significance test on FiSCo being better than SimCSE or SentenceT5?

4. Why does the paper consider suitability: “Is ChatGPT appropriate for answering this question?” Based on the examples in appendix B.1.2., it seems unlikely that the models tested would have come up with \emph{unsuitable} responses to the prompts defined in the appendix (given the specificity).

5. Is asking annotators “Is a human likely to ask this question?” a reasonable substitute to, for example, gathering questions from the WildChat dataset (Zhao et al.)? The latter may have better ecological validity for real-world user queries in the wild.

6. I am a bit confused as to why we would consider the weighted average between entailment, neutral, and contradiction counts in equation (1). Can you provide some intuition about why only focusing on entailment “reflects our goal of identifying shared information between responses without penalizing for contradictions or neutrality in the initial analysis phase”? Is considering all three of these in a single metric something inherited from the hallucination detection literature?

References
Zhao et al. WildChat: 1M ChatGPT Interaction Logs in the Wild.

Comments
1. You may wish to cite that gender, race, and age are protected categories (at least in the US, and probably in many other countries) as defined by the Equal Employment Opportunity Commission.

2. For future work, I believe that the method should easily extend to measuring notions of intersectional group fairness!

**Reasons To Accept:**

The paper focuses on an important issue that has been difficult to quantify, namely, bias in long form generations. I think some related work that would be useful to cite is Eloundou et al., who also focus on fairness in long form generations. The main reasons I see to accept this paper are:

Propose a new way to think about group fairness of long form generations in terms of claims and entailment of these claims between and across groups. I believe this is a significant contribution, since even thinking about what group-fairness means for long-form generations is often difficult. By establishing a quantitative measure, this allows the field to move towards creating fairer LLMs and algorithmic systems. This metric seems generalizable and could be applied to future studies in group-fairness.

Statistical significance of results. It seems that the authors have put careful thought into ensuring that their metric is a statistically significant improvement upon prior proposed metrics in both textual similarity and group fairness. Furthermore, the proposed FiSCo metric has lower variance than the existing ones (e.g. CRougeL, FairPair, or CCosine) due to focusing on entailments of claims, which are more robust among different generations (than syntactic differences which many previous measures rely on).

References
Eloundou et al. First-person fairness in chatbots.

**Reasons To Reject:**

I think the main reasons to reject this paper would be based on the descriptions in the experiments section 5. I had some concerns about this section:

1. Although expanded upon in Appendix A, I believe that the experiment setup section in 5.1 is not very clear. I would appreciate an example figure which shows how the experiment is set up, what the similarity score is measuring, etc. I am also not clear on what the p-values are measuring (more discussion in the questions section of the review).

2. Although it is clear how the prompt templates swap for the “gender” sensitive attribute, I don’t know if the paper clearly explains how “embedding age-related cues in the prompt and comparing responses for younger versus older individuals” (page 6) works. Is this explained somewhere in the appendix?

3. How is agreement calculated in the experiments in section 5.2? The paper just states agreement with the ground truth, how is this exactly calculated given that FiSCo is a metric in [0,1]? Also, if there is only a 70% agreement by FiSCo with the ground truth, how should I interpret a FiSCo score of 0.13, for example, for Claude3 Sonnet?

4. In algorithmic fairness, new fairness notions are often accompanied by explanations of their utility, or what kind of ``world" that they would push generations towards. To some extent, this seems a bit lacking in the paper. Why should I care about the age-wise FiSCo score of Claude 3 for example? At it's core, FiSCo is a way to measure similarities between text. Is there some qualitative takeaway that I can derive from a certain score? I could be asking for too much, since my background is more-so in model calibration (where we have concrete, interpretable takeaways for a certain calibration level).

5. Significance of the results comparing FiSCo to other similarity metrics hinges importantly on the synthetic / human annotated data used to evaluate. Since the paper does not use an open benchmark, the quality of this data is very important. However, the data is not available in the supplementary material.

Regardless, I thank the authors for going through the effort of collecting human annotations for their prompt response templates, which surely helps ground the study.

---

> ### Author Response · Authors · 2025-06-02
> **Metrics calculation**
>
> Thank you for acknowledging our contribution in **group fairness in long-form generations**. Yes, the motivation of this work is to find a systematic way to evaluate LLM application. Thus, your recognition of our **statistical significance of our results** and **low variance** are essential and encouraging. We will cite  Eloundou et al., EEOC and WildCat in related work session. We address your concerns with the following clarifications:
>
> ## How is agreement calculated in the experiments in section 5.2? The paper just states agreement with the ground truth, how is this exactly calculated given that FiSCo is a metric in [0,1]? Also, if there is only a 70% agreement by FiSCo with the ground truth, how should I interpret a FiSCo score of 0.13, for example, for Claude3 Sonnet?
>
> Answer: While the similarity score is bounded in [0,1], we compute a t-statistic over grouped similarity values as described in Section 3.5 (referred to as the FiSCo score). We then apply a significance threshold (α = 0.05) to make a binary decision about whether a given group of texts exhibits statistically significant bias. For agreement calculation, we compare this binary decision (biased or not biased) to our ground truth annotations.
>
> Regarding the interpretation of scores like 0.13 for Claude3 Sonnet: this means that in 13% of cases, we detected statistically significant differences between groups, indicating potential bias. In other words, 13% of evaluated prompt cases were classified as biased.
>
> We will clarify this interpretation in the paper to avoid confusion.
>
> ## Further discussion on the p-values presented is needed. We have a p-value from the Welch’s t-test used to evaluate FiSCo. However, is this the same p-value cited in Table 1? Or do the p-values in table 1 represent a significance test on FiSCo being better than SimCSE or SentenceT5?
>
> Answer: The p-values serve different purposes in different parts of our analysis:
>
> For the FiSCo score itself, we use Welch's t-test to determine whether the differences between inter-group and intra-group similarities are statistically significant. This helps us identify whether a given instance shows evidence of bias.
>
> In Table 1, we compare the performance of different similarity metrics. Here, we use a paired t-test (measuring scores on the same sample across different methods) to assess whether the mean difference in agreement rates between our method and the second-best method (SentenceT5) is statistically significant. We apply the same procedure in Table 2. We will clarify our points in the final version.

---

> > ### Comment · Reviewer_iCei · 2025-06-04
> >
> > Thank you for the detailed feedback on my questions. I will raise my score to a 7. In terms of (optional) suggestions for any future drafts:
> >
> > 1. You can probably drop the contradiction / neutral terms from equation (1) for clarity if you never use them in the main paper. You can refer to the appendix for additional discussion if necessary.
> > 2. Improving the clarity in p-values (as you've already discussed).
> > 3. "Concrete, qualitative examples to demonstrate how certain FiSCo scores can be interpreted in context and used to guide fairness interventions"
> >
> > Thank you!

---

> > > ### Author Response · Authors · 2025-06-10
> > >
> > > 1. agree. That will make the paper simplier. I will change it accordingly.
> > > 2. I will add more details to distinguish p-value in our final draft.
> > > 3. Agree. I will add some examples of qualitative study in the paper.
> > >
> > > Thank you for your amazing feedback!

---

> ### Author Response · Authors · 2025-06-02
> **Practical Implications**
>
> ## In algorithmic fairness, new fairness notions are often accompanied by explanations of their utility, or what kind of ``world" that they would push generations towards. To some extent, this seems a bit lacking in the paper. Why should I care about the age-wise FiSCo score of Claude 3 for example? At it's core, FiSCo is a way to measure similarities between text. Is there some qualitative takeaway that I can derive from a certain score? I could be asking for too much, since my background is more-so in model calibration (where we have concrete, interpretable takeaways for a certain calibration level).
>
> Answer: We appreciate the reviewer’s point about the importance of articulating the real-world utility of fairness metrics. FiSCo is indeed grounded in textual similarity, but its purpose goes beyond measuring linguistic overlap — it offers a quantifiable signal of differential treatment in generated outputs across social groups, under controlled conditions.
>
> This becomes particularly important when generation tasks have potential downstream consequences. For example, a high gender-based FiSCo score indicating notable differences in career advice — even with all other variables held constant — suggests that an LLM may be reinforcing or amplifying existing societal disparities in professional guidance. This isn’t simply a matter of stylistic variation; it could affect users’ long-term decisions and opportunities in ways that are unjustified or inequitable.
>
> Similarly, elevated age-based FiSCo scores (e.g., for Claude 3) may point to systematic shifts in tone or substance when addressing older vs. younger individuals. In contexts like hiring, mentorship, or performance evaluations, such behavior could contribute to age discrimination — whether intentional or not.
>
> While we acknowledge that FiSCo is not a fairness solution in itself, it is a diagnostic tool — one that highlights where disparities exist so that practitioners can investigate further and take appropriate action. By surfacing group-level response differentials, FiSCo provides actionable insights that can inform model development, auditing, and mitigation strategies. In this way, FiSCo contributes to the broader goal of pushing generations toward a world that does not encode or perpetuate unjust treatment of specific groups. Analogous to how calibration scores signal over- or under-confidence, FiSCo flags disparities that merit closer human inspection.
>
> In the final version of the paper, we will be clearer to articulate these implications and provide concrete, qualitative examples to demonstrate how certain FiSCo scores can be interpreted in context and used to guide fairness interventions.
>
> ## I am a bit confused as to why we would consider the weighted average between entailment, neutral, and contradiction counts in equation (1). Can you provide some intuition about why only focusing on entailment “reflects our goal of identifying shared information between responses without penalizing for contradictions or neutrality in the initial analysis phase”? Is considering all three of these in a single metric something inherited from the hallucination detection literature?
>
> Answer: You're right — our metric is indeed inspired by and inherits design choices from the hallucination detection literature.
>
> We focus specifically on entailment (by setting α = 1 and the weights for neutral and contradiction to 0) in applications where differences in content could lead to long-term impacts on certain groups. For example, when using an LLM to assess candidate strengths for a job, even subtle variations in the way different groups are described may influence hiring decisions and affect both individuals and institutions over time.
>
> By prioritizing entailment, we emphasize shared information between responses, which allows us to detect whether responses for different demographic groups omit key aligned content. A lack of shared entailments can be an indicator of bias, particularly when group responses diverge in substance rather than just style.
>
> You're also correct that combining entailment, neutral, and contradiction into a single score is a common practice in hallucination detection. However, in our bias detection setting, we found that focusing solely on entailment produced clearer and more actionable signals, especially in the initial analysis phase.
>
> We will add this explanation in the final version to clarify our design choice in Equation (1).

---

> ### Author Response · Authors · 2025-06-02
> **Data Availability, Age bias definition, figures, and other suggestion**
>
> You raise an excellent point about the importance of synthetic dataset quality. We apologize for not including our sampled dataset in the supplementary materials. We will share the dataset upon the final decision of the paper to support a thorough review of question quality and to enable replication of our results.
>
> To operationalize age as a bias variable, we included a statement such as "I am XX years old" in the prompt, while keeping all other contextual attributes the same (see Table 6). In our setup, we defined individuals above age 50 as ‘older’ and those between 20 and 50 as ‘younger’, and we generated an equal number of examples for both groups. We will add these details explicitly in the final version.
>
> Regarding the figure suggestion: repositioning is a great idea. We initially planned to show the t-SNE figure in the introduction. However, we felt that explaining baseline comparison methods and the meaning of embeddings in the t-SNE space might overwhelm readers early in the paper. Therefore, we opted to present a simple illustrative example as Figure 1 — one that operates in the discrete output space rather than the embedded space.
>
> Finally, we will cite the EEOC guidelines as suggested, and in future work, we plan to extend our analysis to consider intersectional group fairness metrics. For statistical testing across multiple subgroups, we expect to use one-way ANOVA to test for mean differences in FiSCo scores across intersectional groupings.

---

> ### Author Response · Authors · 2025-06-02
> **Human Evaluation**
>
> ## Why does the paper consider suitability: “Is ChatGPT appropriate for answering this question?” Based on the examples in appendix B.1.2., it seems unlikely that the models tested would have come up with \emph{unsuitable} responses to the prompts defined in the appendix (given the specificity).
>
> Answer: We performed the validation procedure to ensure that the questions themselves (not the responses) were a good representation of realistic use cases that LLMs would be expected to handle. As noted in Appendix B, we identified two types of unsuitable questions:
> (1) those that violate ethical, moral, or legal standards, and
> (2) those that are improbable, incoherent, or meaningless.
>
> LLMs may still generate answers to such prompts, but these questions do not reflect realistic or appropriate user–LLM interactions. Evaluating model behavior on such inputs would dilute the signal we aim to measure with FiSCo.
>
> Additionally, unspecific questions lacking necessary background information can also be unsuitable, as they do not allow for meaningful or grounded comparisons. This is illustrated in Table 9 (row 3), where the lack of context renders the prompt ineffective.
>
> ## Is asking annotators “Is a human likely to ask this question?” a reasonable substitute to, for example, gathering questions from the WildChat dataset (Zhao et al.)? The latter may have better ecological validity for real-world user queries in the wild.
> We synthesized questions that were likely to elicit bias-related behaviors from LLMs. To do this, we first adopted the Advice and Insight Generation templates to generate question topics. We then augmented these templates with demographic attributes (e.g., gender, race, age) as potential bias triggers.
>
> Answer: After adding these attributes, we conducted a likelihood validation step, where annotators judged whether a human would realistically ask each question. This helped ensure that the final prompts were not only diverse and bias-relevant but also realistic and contextually plausible. As shown in Table 9 (row 1), we excluded questions where personal information was irrelevant to the query — i.e., prompts that no real user would plausibly ask.
>
> We agree that the WildChat dataset is a valuable resource for collecting real-world user–LLM interactions and may serve as a strong starting point for question topic inspiration. However, once bias-triggering factors are inserted, human validation is still essential to ensure the resulting questions remain ecologically valid and contextually grounded. In our work, this two-stage process (template-driven synthesis + annotator validation) was necessary to maintain both bias relevance and user realism.

---

### Official Review · Reviewer_DXZz · 2025-05-21

**Rating:** 7
**Confidence:** 4
**Ethics Flag:** 1

**Summary:**

This paper proposes a framework and methodology for assessing bias in open-ended text generation. The key idea behind the framework is a new group fairness metric that captures the (dis)similarity in outputs generated by counterfactual prompts (prompts that are different only in the target bias group). The proposed method studies the (dis)similarity within a particular group (intra-group) and also between groups (inter-group). The new group fairness metric is validated on both synthetically generated data and human-annotated data.

The similarity between two LLM responses is computed by first splitting the generated text into claims and then using NLI/entailment relations to compute the similarity between two generated responses, where entailment corresponds to most similar and contradiction to the least similar. Intuitively, the more dissimilar the responses to counterfactuals, the higher the bias of the model.

To showcase the new group fairness metric, the paper introduces a template-based dataset that focuses on contexts that could lead to differential treatment. The target bias groups are gender, race and age. Several state-of-the-art models are assessed for bias, with Claude Sonet showing the lower bias score.

**Questions To Authors:**

I really enjoyed reading the paper. I appreciate the formulation of the new group fairness. I find the (dis)similarity between the generated output as a proxy for bias intriguing. Using counterfactuals to assess bias reminded me of the following datasets/works: BBNLI https://github.com/feyzaakyurek/bbnli and BBNLI-next https://arxiv.org/abs/2305.12620. In particular, the BBNLI-next discusses how some mispredictions (as proxy for bias) could be due to the lack of the model robustness and proposes disaggregating bias measures.

I have a few clarifying questions.

* If I look at the second example in Table 13, I’m not sure I would categorize the difference between Text1 and Text2 as bias. In fact, I think there could be instances where the differences are ok and not a proxy for bias. In particular, for the example in Table 13, I don’t find the difference as indicative of bias. Do you think that different model behavior for counterfactuals is always indicative of bias?

* Some of the templates could lead to different generated output and I wouldn’t find it problematic/indicative of bias. For example, wrt this template: “Describe [NAME]’s communication style with team members.” may lead to different answers for women/men that are not problematic; the same for different ages. In other words, it is not clear to me how to distinguish between differences that are indicative of bias/problematic behavior/differential treatment, and the ones that are benign.

In the first paragraph in 4.3, it’s not clear to me what (r1, r2, r3) refer to (including line numbers makes it easier to provide specific line feedback)

The paper is well written. I like that it tries to educate the reader wrt existing problems, datasets, etc. I suggest including in the discussion the following papers/works:

Problems with metrics and datasets:

Intrinsic Bias Metrics Do Not Correlate with Application Bias
https://aclanthology.org/2021.acl-long.150/

Why Don’t Prompt-Based Fairness Metrics Correlate?
https://aclanthology.org/2024.acl-long.487/

Stereotyping Norwegian Salmon: An Inventory of Pitfalls in Fairness Benchmark Datasets
https://aclanthology.org/2021.acl-long.81/


Datasets that go beyond race, gender, age, etc.:

Bias Against 93 Stigmatized Groups in Masked Language Models and Downstream Sentiment Classification Tasks
https://dl.acm.org/doi/10.1145/3593013.3594109

SocialStigmaQA: A Benchmark to Uncover Stigma Amplification in Generative Language Models
https://arxiv.org/abs/2312.07492

Benchmarks for assessing bias:

CALM : A Multi-task Benchmark for Comprehensive Assessment of Language Model Bias
https://arxiv.org/abs/2308.12539

**Reasons To Accept:**

The proposed framework is a step in the right direction for assessing social bias in open-ended text generation.
The paper is well written and motivated.

**Reasons To Reject:**

A few details could be better explained, nothing major.

---

> ### Author Response · Authors · 2025-06-02
>
> Thank you for believing **our proposed framework is a step in the right direction for assessing social bias in open-ended text generation.** We will include BBNLI and other suggested works in our paper. We address your concerns with the following clarifications:
>
> **1** If I look at the second example in Table 13, I’m not sure I would categorize the difference between Text1 and Text2 as bias. In fact, I think there could be instances where the differences are ok and not a proxy for bias. In particular, for the example in Table 13, I don’t find the difference as indicative of bias. Do you think that different model behavior for counterfactuals is always indicative of bias?
>
> Answer: Thank you for going through the Appendix and for this thoughtful observation about Table 13. You raise an important point - differences in model outputs don't automatically constitute bias. Whether a difference indicates bias depends heavily on context and potential impact. In the example you referred to, while the responses themselves may appear neutral, suggesting a technical career path (computer science) for the female engineer versus a management track for the equally qualified male engineer could potentially reflect and reinforce existing gender disparities in technical leadership roles. Therefore, this difference could be indicative of bias, particularly when the question concerns career planning. Different model behavior for counterfactuals is not always indicative of bias — it strongly depends on the specific context and the nature of the question. Thus, we dedicated section 4.1 for identifying important questions.
>
> **2**. Some of the templates could lead to different generated output and I wouldn’t find it problematic/indicative of bias. For example, wrt this template: “Describe [NAME]’s communication style with team members.” may lead to different answers for women/men that are not problematic; the same for different ages. In other words, it is not clear to me how to distinguish between differences that are indicative of bias/problematic behavior/differential treatment, and the ones that are benign.
>
> Answer:  You're absolutely right to point out that not all differences across demographic groups are inherently indicative of bias. For example, variations in how communication styles are described across gender or age may reflect genuine diversity in expression and social norms, and these differences are not necessarily problematic.
>
> However, our analysis goes beyond observing individual template outputs. As discussed in Table 5, the base prompts lack sufficient contextual grounding to enable reliable bias assessment on their own. To address this, we augmented the questions with additional controlled attributes (see Table 8), such as experience, role, and qualifications. This augmentation enables more controlled and meaningful group comparisons.
>
> When we observe systematic differences in how groups are described — after controlling for all relevant variables — we argue that such patterns can signal bias, especially when those differences are likely to influence real-world outcomes, such as hiring, promotions, or performance evaluations.
>
> That said, we agree this is a nuanced issue. Bias is not simply a matter of detecting any difference — it requires careful interpretation of context, potential harm, and intention. Our goal is not to flag every disparity as harmful, but rather to identify where unjustified or disadvantageous differentials may exist, and to enable more transparent evaluation of those effects.
>
>
> **3**. In the first paragraph in 4.3, it’s not clear to me what (r1, r2, r3) refer to (including line numbers makes it easier to provide specific line feedback)
>
> Answer: While we mention the definition in Section 3.3, we agree that the notation (r1, r2, r3) could be clarified further. These symbols refer to different responses generated for the same question. We will revise Section 4.3 in the final version to make this notation more explicit and easier to follow for the reader.

---

> > ### Comment · Reviewer_DXZz · 2025-06-04
> >
> > This nuanced discussion should go in the paper. If you have time and resources, it would be nice to include a qualitative analysis of some - let's say 100 pairs - generated pairs randomly selected to better understand how often discrepancy is indicative of bias.

---

> > > ### Author Response · Authors · 2025-06-10
> > >
> > > Thank you for insightful feedback. We will try to add discussion in the paper. I can provide some more concrete examples of indicative of bias in our final draft.

---

> > > > ### Comment · Reviewer_DXZz · 2025-06-10
> > > >
> > > > Best of luck incorporating all the feedback/suggestions! I loved the paper and I wish I could share it already with others :)
> > > >
> > > > PS: Manually looking at the data (a small sample) and incorporating the analysis (even in the appendix if not in the main body of the paper) sends a (strong?) signal that we don't only rely on overall statistics/automatic evaluation, but also need to look at the *actual* data and keep the human in the loop. The problems you are addressing are socio-technical problems, we can't keep the human outside of the loop =)

---

### Decision · Program_Chairs · 2025-07-08

**Decision:**

Accept

**Comment:**

This paper proposes a new way to think about group fairness for generative models with long-form and open-ended response generation by using counterfactual prompts and breaking the responses into claims and further doing statistical analysis on the entailments of claims in various responses belonging to the different sensitive groups. All reviewers appreciated the importance of the problem, the proposed framework taking a step in the right direction for assessing fairness in text generation, the writing of the paper, as well as extensive results with proper statistical analysis. The proposed metric has been shown to be clearly better than existing metrics in the literature. Overall, this is a great paper that should be accepted to the conference. Congratulations to the authors!